# N-Doped Graphene-like Film/Silicon Structures as Micro-Capacitor Electrodes

**DOI:** 10.3390/ma16114007

**Published:** 2023-05-26

**Authors:** Daria M. Sedlovets

**Affiliations:** Institute of Microelectronics Technology and High-Purity Materials, Russian Academy of Science (IMT RAS), Moscow District, 6 Academician Ossipyan Str., 142432 Chernogolovka, Russia; sedlovets@iptm.ru; Tel.: +7-4965244190

**Keywords:** on-chip devices, graphene-like film, nitrogen doping, electrochemical capacitance, voltammetry, impedance spectroscopy

## Abstract

Currently, the miniaturization of portable and autonomous devices is challenging for modern electronics. Graphene-based materials have recently emerged as one of the ideal candidates for supercapacitor electrodes, while Si is a common platform for direct component-on-chip integration. We have proposed the direct liquid-based CVD of N-doped graphene-like films (N-GLFs) on Si as a promising way to achieve solid-state on-chip micro-capacitor performance. Synthesis temperatures in the range from 800 °C to 1000 °C are investigated. Capacitances and electrochemical stability of the films are evaluated using cyclic voltammetry, as well as galvanostatic measurements and electrochemical impedance spectroscopy in 0.5 M Na_2_SO_4_. We have shown that N-doping is an efficient way to improve the N-GLF capacitance. 900 °C is the optimal temperature for the N-GLF synthesis with the best electrochemical properties. The capacitance rises with increasing film thickness which also has an optimum (about 50 nm). The transfer-free acetonitrile-based CVD on Si yields a perfect material for microcapacitor electrodes. Our best value of the area-normalized capacitance (960 mF/cm^2^) exceeds the world’s achievements among thin graphene-based films. The main advantages of the proposed approach are the direct on-chip performance of the energy storage component and high cyclic stability.

## 1. Introduction

Graphene has recently emerged as one of the ideal candidates for electrode materials for supercapacitors with excellent theoretical capacitance [1,2,3]. Heteroatom doping is a potential approach to significantly improve the energy storage performance of graphene materials by introducing the defects and increasing the electrode conductivity [4,5,6]. Nitrogen is widely used as a dopant since it can enhance the number of charge carriers by adding its p electrons to the graphene π-system [7]. A large number of surface defects are also induced into the graphene films by N-doping, which increase the reversible discharge capacitance [8]. As reported in work [9], nitrogen-doped graphene nanoplatelets (NGN) exhibit improved specific capacitance due to enhanced electrical conductivity and Faradic redox activity. The former can contribute to electrical double-layer (EDL) capacitance, and the latter can increase pseudocapacitance. Nitro functionalities enhance the pseudocapacitance, increasing the active surface area accessible to an electrolyte by improving the wettability of the electrodes [10,11]. It has been reported that nitrogen-doping levels in the NGN can play important roles in determining their specific capacitance [9]. The authors [12] have found an increasing capacitance of graphene synthesized through liquid-based CVD on Cu with increasing N content. On the other hand, nitrogen-doped reduced graphene oxide (N-RGO) showcases superior capacitive behavior (280 F/g without capacity loss after 40,000 cycles in 1M H_2_SO_4_), even though the nitrogen amount is only 0.48% [13]. An efficient EDL formation can be attributed to the unique porous structure and superior conductivity of the N-RGO electrode. Close capacitance values (282 F/g) have been reported for N-RGO (about 2% N content) in 6M KOH [14]. Not only the content but also the type (bonding configurations) of nitrogen is important. There are three major types of N heteroatom bonding configurations within the carbon lattice: sp^3^ pyrrolic (1), sp^2^ pyridinic (2) edge states and quaternary/graphitic, and (3) N attributed to atoms substituting carbon in the benzene ring. Pyrrolic N induces more structural defects than graphitic [15]. Pyridinic/pyrrolic groups can increase the total energy density as they get involved in reversible redox reactions, while graphitic N improves the electrochemical performance by enhancing conductivity [7].

Despite advancements in graphene technology and related materials, scalable and inexpensive preparation methods must be further developed [10]. For successful practical application, novel synthesis techniques should be exploited, so the N-doped graphene can be produced directly on non-metal substrates, eliminating the process of transferring the film from the foil surface. Moreover, direct synthesis allows obtaining the graphene films on silicon plates. At present, Si is a common semiconductor material that can be used as a platform for direct component-on-chip integration toward the miniaturization of portable and autonomous electronic devices.

Only a few groups have attempted direct transfer-free chemical vapor deposition (CVD) on non-metal substrates: Si [16], SiO_2_ [17,18,19], glass [20], and sapphire [19]. Recently, we have proposed a simple and efficient approach for the transfer-free CVD synthesis of N-doped graphene-like films (N-GLFs) on non-metals through pyrolysis of acetonitrile. Here, the electrochemical characteristics of the N-GLFs directly synthesized on silicon are investigated, and their energy storage behavior is described concerning the bonding configurations of nitrogen and thickness of the N-GLFs. Our micro-capacitor (μC) electrodes have better characteristics than other reported results.

## 2. Materials and Methods

### 2.1. N-GLF Deposition

The single-crystal arsenic-doped n-type silicon (0.001–0.003 Ω∙cm) plates (Telecom-STV, Zelenograd, Russia) with a surface orientation (100) were used as wafers. After cutting with a diamond pencil, the samples were sonicated in isopropanol (99.8% 2-C_3_H_7_OH (EKOS-1, Moscow, Russia)).

CVD of the N-GLFs was carried out through high-temperature (800–1000 °C) low-pressure (~100 Pa) pyrolysis of acetonitrile (99.9% CH_3_CN (Cryochrom, St. Petersburg, Russia)) vapor in a carrier gas flow (99.999% Ar (Linde Gas, Balashikha, Russia)). N-GLF growth was carried out in a horizontal quartz reactor 1 (of 50 cm length and 3 cm inner diameter) incorporated in a tubular furnace 2 with a temperature control system: SPC-1-50 power unit (Autonics, Busan, Republic of Korea) and TZN-4S controller (Autonics, South Korea). The inlet and outlet of the reactor were linked to the reagent supply system and forevacuum pump VI-2120 (Value, Taizhou, China), respectively. The liquid precursor acetonitrile (99.9% CH_3_CN (Cryochrom, St. Petersburg, Russia)) from the graduated burette was fed through a peristaltic pump Sci Q400 (Watson Marlow, Harpen, UK) into the reactor where it was vaporized. The carrier gas flow (99.999% Ar (Linde Gas, Balashikha, Russia)) was maintained with gas flow regulator RRG-10 (Eltochpribor, Moscow, Russia). The working pressure (1 kPa) was measured with a vacuum gauge DVR2 (Vacuubrand, Wertheim, Germany). For reference sample deposition (non-doped GLFs), ethanol was used as a liquid precursor. The CVD set-up scheme and synthesis diagram were given in Appendix A.

After evacuation, the CVD set-up was heated to the required temperature for 10 min in argon (2 L/h) acetonitrile (2 mL/h) flows. Then the flow rate of Ar and CH_3_CN was increased to 4 L/h and 4 mL/h, respectively. Synthesis time was varied in the range of 0.5–54 h. At the end of the deposition process, the system was cooled down to room temperature via inert gas purging. Synthesis conditions are summarized in Appendix A).

### 2.2. Measurements

Before measurements, the sample was placed in an electrolyte overnight for wetting, and several cycles were preliminarily carried out for activation according to previous observations [21]. Weak and noisy CV curves before soaking in electrolytes are shown in Appendix A. Electrochemical measurements were performed on a computer-controlled potentiostat P-40X in a three-electrode cell E-6S (Electrochemical Instruments, Chernogolovka, Russia) with an Ag/AgCl reference electrode and a graphite counter one. N-GLF synthesized on Si was used as the working electrode (electrical contact was performed through the silicon substrate). The measurements were carried out in an aqueous electrolyte (0.5 M Na_2_SO_4_) ranging from −800 mV to 800 mV. When the cyclic voltammograms (CVs) were recorded, the scan rates were varied from 100 mV/s to 5 mV/s. Galvanostatic charge-discharge (GCD) measurements were performed at a current density of 0.05 mA/cm^2^. Electrochemical impedance spectra (EIS) were recorded in the 50 kHz–10 Hz interval.

The specific capacitance was calculated by integrating the area under the CV curve to obtain the charge value and, then, dividing this by the surface area, the scan rate, and the potential window, according to Equation (1) [22]:(1)$$C=\frac{\int IdV}{∆V\cdot \vartheta \cdot A}$$
where C is the area-normalized capacitance in F/cm^2^, ∫IdV is the integrated area of the CV curve, ΔV is the scanned potential window in V, ϑ is the scan rate in V/s, A is the surface area which was exposed to the electrolyte, in cm^2^.

### 2.3. Characterization

To determine the film thickness, the sample was scratched, and the height difference at the N-CLF/Si interface was measured using an Integra atomic force microscope (AFM).

We also refer to the results of our previous analytical studies [23]. Raman mapping was done with a Senterra micro-Raman system (Bruker, Berlin, Germany) in the 1000–3600 cm^−1^ range under 0.5 cm^−1^ resolution using a 532 nm laser. A total of 24 spectra were recorded for each sample at different points. During the quantitative analysis of the spectral data, the baseline subtraction was calculated, and the peak parameters were extracted to determine the ratios of D and G band intensities as the structural perfection estimation.

X-ray photoelectron spectroscopy (XPS) data were obtained using PHOIBOS 150 MCD spectrometer (Specs, Berlin, Germany) at 3 × 10^−10^ Torr. An X-ray tube with magnesium anode (Mg K_α_ = 1253.6 eV) was used as a source with a power of 225 W. The spectra were recorded in the constant transmission energy mode (40 eV with 1 eV step for the survey spectrum and 10 eV with 0.03 eV step for individual lines). Energy calibration of XPS was made using the Ag 3 d_5/2_ = 368.27 eV peak.

## 3. Results and Discussions

CVs were recorded at varied scan rates (see Appendix A). Then the area normalized capacitances were calculated from CV curves as ascribed in Section 2.2 for N-doped samples synthesized at 800–1000 °C. To estimate the effect of synthesis time on the capacitance of deposited materials, 2 series of samples were synthesized: “thin” (Figure 1a,c) and “thick” (Figure 1b,d) films with sheet resistances of about 100 kOhm/square and 10 kOhm/square, respectively. During the measurements, it was observed that the capacitance value changes noticeably after cycling. Because cyclic stability is no less an important characteristic than capacitance value, we presented the measurement results before and after 5000 cycles.

Figure 1a shows that all thin samples have close capacitance values, regardless of the synthesis temperature. As seen in Figure 1b, the absolute values of the capacitance for thick N-GLFs increase, while the temperature dependence exhibits the same behavior as for thin films. Both thin and thick samples synthesized at 1000 °C have the lowest capacitance at a minimal scan rate. After a short cycling (see Figure 1c), thin N-GLF synthesized at 900 °C has the highest capacitance (about 90% retention of initial value at 5 mV/s). For other samples, the capacitances decrease significantly, especially for the film synthesized at 800 °C (70% capacitance loss). After 5000 cycles (see Figure 1d), a loss of capacitance is observed just for thick film synthesized at 800°C, while for the samples synthesized at 900 °C and 1000 °C, a slight increase in capacitance appears. This is probably due to an improvement in the wettability of the material by the electrolyte during cycling, as also described in [24]. Notably, after cycling, the capacitance of the N-GLF synthesized at 1000 °C is less than at 900 °C for both sample series (thin and thick films).

From the results above, we summarized that 900 °C is the optimal temperature for synthesizing the N-GLFs with the best electrochemical properties. 800 °C yields in the least stable material; 1000 °C—in the samples with the lowest initial capacitance at a minimal scan rate and lower capacitance (compared to 900 °C) after cycling. To explain the effect of the synthesis temperature on the electrochemical properties of the synthesized N-GLFs, we consider two temperature-dependent characteristics (crystallite size and nitrogen configurations), which we investigated recently [23].

From Raman spectroscopy, it follows that with an increase in the synthesis temperature, the crystallite size of the deposited film increases (see Appendix A or ref. [23]). Considering the smallest crystallite size in the N-GLF deposited at 800 °C, it can be assumed that this material gets structural disturbance due to the facilitated segregation of the graphene flakes from the solid film under the influence of the electrolyte and electric current. An extreme capacitance loss in the thin N-GLF deposited at 800 °C can be caused by damage to the film continuity through the breaking off of the smallest flakes during the cycling measurements. With a longer deposition time (thick films), the lateral interactions between graphene grains in the film are enhanced, and the capacitive losses become less noticeable. Possibly the lowest synthesis temperature also provides the poorest adhesion.

As discussed in the Introduction, the bonding configurations of nitrogen significantly affect the material’s properties. We have earlier found [23] that N-type in the N-GLFs depends on the temperature of their synthesis (see Table 1).

At the lower temperature (800–900 °C) the main type of nitrogen is graphitic, but pyrrolic and pyridinic are also contained, while at 1000 °C, almost all nitrogen atoms are replaced by carbon in the lattice, practically without forming pyrrolic and pyridinic bonds. So, low electrochemical characteristics of the N-GLF deposited at 1000 °C may arise from extremely low concentrations of pyrrolic and pyridinic N configurations contained in the N-GLFs. It is these configurations that create structural defects functioning as active sites in electrochemical processes. Our films themselves are quite defective due to their nanocrystalline structure. So, the efficient electrochemical performance of the N-GLFs does not require a high content of pyridinic and pyrrolic bonds, but their almost complete absence can cause a decrease in capacitance.

The best characteristics were achieved for the N-GLFs synthesized at 900 °C. Importantly, the obtained capacitance values were close to that calculated from the GCD data (see Appendix A). Film thickness growth, as well as cycling, increased the capacitance. Therefore, we continued investigating prolonged cycling (see Figure 2a) and further increased the synthesis time at the same temperature (see Figure 2b). Each sample was subjected to cycle measurements (at 1000 mV/s), and the scan rate dependence of the capacitance was determined after every 5000 cycles.

As seen in Figure 2a, the capacitance value continued to grow during prolonged cycling of the sample synthesized at 900 °C for 26 h. After 20,000 cycles, as a result of an additional increase in capacitance, the maximum value of 960 μF/cm^2^ was achieved (at 5 mV/s). After the longer synthesis time (35 h, see Figure 2b), the initial capacitance increased, but its additional growth during cycling occurred only up to 10,000 cycles. After 20,000 cycles, the capacitance dropped slightly, and the maximal absolute value was lower than that for the sample synthesized for 26 h.

Therefore, it can be concluded that the synthesis time (and, accordingly, the film thickness) has an optimum at which no further increase in the capacitance is observed. Probably the films become too thick, causing difficulties in their interaction with the electrolyte. EIS measurements confirm the presence of diffuse hindrances in the thicker sample. See Figure 3a: the Nyquist plot for the 35 h has a higher slope than that of the 26 h. This means that the reactance of the 35 h sample is increased compared to its active resistance that can arise from the worsened electrical contact.

Finally, we compare capacitive characteristics between N-doped and pristine GLFs. For this purpose, the non-doped GLFs were deposited from ethanol (instead of acetonitrile) vapor through the same liquid-based CVD at maximal temperature (1000 °C). The synthesis time (0.5 h) was chosen to be equivalent to thin N-GLF. As seen in Figure 3b, the latter had a capacitance of two to three times higher than the former. This result led to the conclusion that N-doping is an efficient way to improve the capacitance of the GLFs.

Interestingly, in both cases, the energy storage behavior mainly arises not from the EDL mechanism but due to pseudocapacitance. This becomes apparent from both CV curve shapes (far from rectangular) and the non-linear relationship between capacitance and scan rate. The latter originates from incomplete redox reactions at elevated scan rates [25]. The disadvantage of pseudocapacitors is their short lifetime, which is often limited to <10,000 cycles because the redox reactions are not fully reversible and also accompany the expansion-constrain of electrodes [14]. We attribute our successful results (excellent stability after 20,000 cycles) to the good adhesion of the N-GLFs to silicon, achieved in a direct CVD process that does not require post-synthesis transfer procedures.

## 4. Results Evaluating

Miniaturization of portable equipment is an actual challenge for modern electronics. Micro-electrochemical capacitors possess scaling down and on-chip integration energy storage components [26]. Direct integration of the μCs with other elements on the same chip assumes surface area limitations; thus, the proposed materials should be compared by the area-normalized capacitances measured in F/cm^2^. In addition, the evaluation of gravimetric capacitance (specific value per unit gram) of thin graphenic films is hindered since their extremely low masses make accurate mass measurements unreliable. To be honest and precise, we have recalculated our best area-normalized values in gravimetric data based on graphite density. Using AFM, the film thickness is determined to be 50 nm (see Appendix A). The surface area of the electrochemically measured sample is 0.9 cm^2^. Therefore, the volume of the active material:$$V=50\;nm\times 0.9\;{cm}^{2}=\left(5\times 10^{-6}\;cm\right)\times 0.9\;{cm}^{2}=4.5\times 10^{-6}\;{cm}^{3}$$

Graphite density ρ = 2.33 g/cm^3^. Therefore, the volume of the active material:$$m=\rho \times V=2.33\;g/{cm}^{3}\times 4.5\times 10^{-6}\;{cm}^{3}=10.5\times 10^{-6}\;g$$

So, gravimetric capacitance can be calculated from the area-normalized one as follows:$$0.96\;mF/{cm}^{2}\times 0.9\;{cm}^{2}\times \left(\frac{1}{10.5\times 10^{-6}g}\right)=8\times 10^{4}\;mF/g=80\;F/g$$

Obviously, we cannot directly compare such theoretically calculated values with the data obtained for the weighed mass of the active material. These calculations are provided here only to show that the gravimetric capacity of our material corresponds to the world value level, which varies from several tens of F/g [27] to several hundreds of F/g [9,13,14].

The area-normalized capacitance data reported in references [28,29,30,31,32,33,34,35,36,37,38,39,40,41,42,43,44,45,46,47,48] for graphene-based materials are summarized in Table 2. The table contains almost no data for N-doped materials. N-graphenic coatings are widely used in electrochemical μC applications, but most authors describe bulk-like material. We do not consider here works that report the gravimetric capacitances of sputtered graphene powders [9,13,14].

Rows in the table are listed in increasing order of the capacitance. As seen, the main reported values are hundreds of μF/cm^2^ [30,31,32,33,34,35,36,37] and units of mF/cm^2^ [38,39,40,41,42,43,44,45]. Micro supercapacitor applications imply micro capacitive operating. The capacitances equal to the value measured in μF/cm^2^ are sufficient for such performance. However, the efficiency of energy devices can be improved. For example, the last two remarkable results [47,48] are achieved by the composite material producing (rGO with additional polypyrrole or manganese oxide). In both cases, the electrodes are prepared by pressing the active material into a paper-like structure. Undoubtedly, these are outstanding results for portable electronic technology that provokes the progress of wearables. However, this approach is not suitable for on-chip applications. In this regard, ”inkjet printing” technologies look most promising if they shift towards the thickness decrease of the obtained film. Sub-μm thickness [36,42] of the active materials makes them unsuitable for on-chip integration on silicon crystals. Notably, all results superior to our achievements are obtained for several microns (or even tens μm) thick films. So, our result (0.96 mF/cm^2^) is the best value for μC on Si application. This value can be increased when the material is used only as a cathode (when measured just in the negative potential window, the capacitance value is about 1.2 mF/cm^2^).

A high value of specific capacitance is not the only goal on the way to perfect electrode performance. Another important characteristic of μC electrodes is their cyclic life. Capacitance retention after 2000 cycles [46] is not a sufficient indicator of electrode stability since an active material has to withstand many more charge-discharge cycles. A much more convincing result is capacity saving after 10,000 [38], 30,000 [41], and even 150,000 [39].

The work on further improvement of electrochemical properties is very hopeful, especially towards N-GLF surface modification (oxidation or deposition of a supplemental active coating), since oxygen-containing surface groups, as well as a layer of transition metal oxides (MnO_2_ or Fe_2_O_3_, for example), can be involved in redox reactions resulting in an additional pseudocapacitance. Research focused on finding the optimal electrolyte (acidic, alkaline, or organic) also appears highly promising.

## 5. Conclusions

By comparing the electrochemical properties of GLFs deposited from ethanol and acetonitrile vapors, it has been demonstrated that N-doping significantly improves the capacitive characteristics of the GLFs. The dependence of the electrode performance on the synthesis conditions was studied. In the investigated range, the synthesis temperature of 900 °C provides the best results. The optimal thickness of N-GLFs is found to be 50 nm. It has been observed that the type of nitrogen bonds does not strongly affect the specific capacitance; however, the absence of pyrrolic/pyridinic groups can lead to reduced capacitance values.

In summary, we have shown that the direct synthesis of the N-GLFs on silicon plates is a promising way to implement the planar on-chip μCs. Our results demonstrate that the transfer-free acetonitrile-based CVD yields a perfect material for μC electrodes. The best value (960 mF/cm^2^ given 180% retention after 20,000 cycles) exceeds the world level of achievements in this field. Moreover, modification of the films (such as oxidation or an additional deposition of transition metal oxides) can further improve the results achieved. Although CVD at 1000 °C is less efficient than at 900 °C, a further increase in the synthesis temperature is also of great interest. Both structural features (nitrogen bond configuration) and capacitive properties of the N-GLFs deposited from acetonitrile at over 1000 °C are still exciting subjects of study in the future.

## Figures and Tables

**Figure 1 materials-16-04007-f001:**
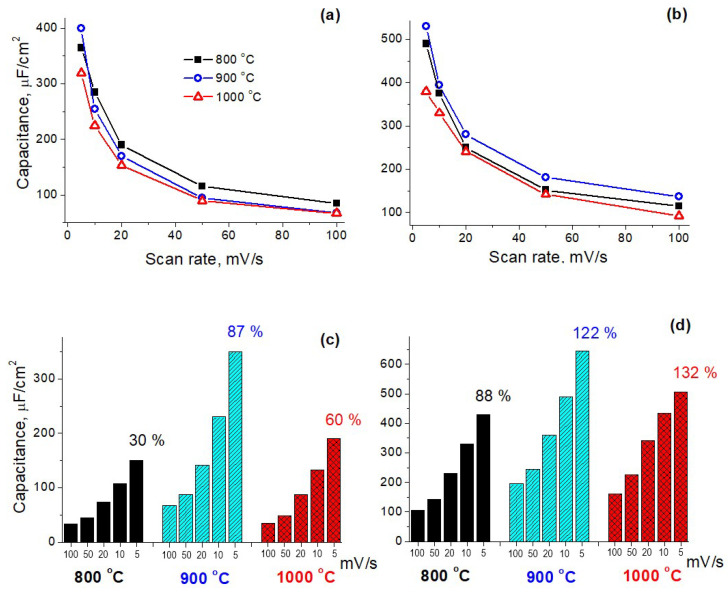
Scan rate dependences of the capacitance values (initial (**a**,**b**) and after 5000 cycles (**c**,**d**) for thin (**a**,**c**) and thick (**b**,**d**) N-GLFs synthesized at 800 °C (black, squares/solid fill), 900 °C (blue, circles/slanting shading) and 1000 °C (red, triangles/rhombic shading). (**b**,**d**) also show the percentage of the capacitance retention after 5000 cycles (data are given for a scan rate of 5 mV/s).

**Figure 2 materials-16-04007-f002:**
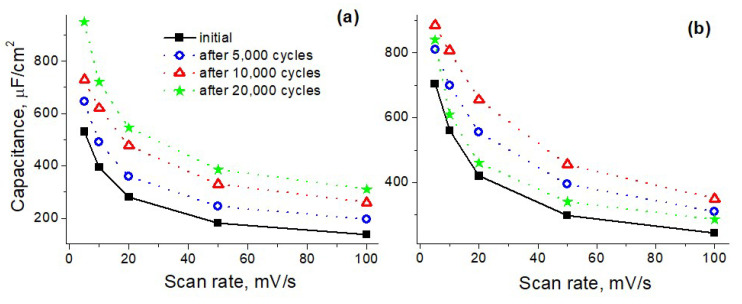
Scan rate dependence of the capacitance values for the N-GLFs synthesized at 900 °C for 26 h (**a**) and 35 h (**b**); initial (black squares) and after 5000 cycles (blue circles), 10,000 cycles (red triangles), 20,000 cycles (green asterisks).

**Figure 3 materials-16-04007-f003:**
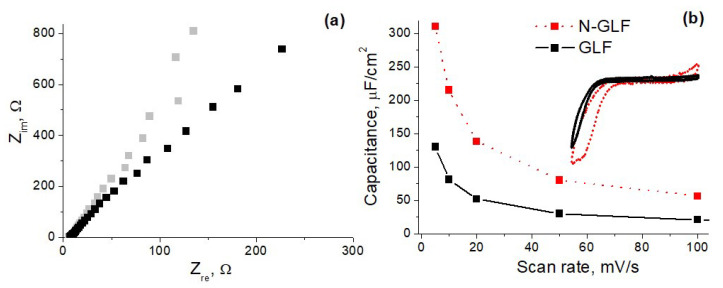
Nyquist plot for the N-GLFs synthesized at 900 °C for 26 h (gray) and 35 h (black) (**a**) and scan rate dependence of the capacitance values (**b**) for N-doped (dotted red) and pristine (black) GLFs synthesized at 1000 °C from acetonitrile and ethanol vapors, respectively. Inset: CV curves recorded at 100 mV/s for N-doped (dotted red) and pristine (black) GLFs.

**Table 1 materials-16-04007-t001:** ^1^ Doping level (nitrogen to carbon ratio) and the relative content of nitrogen atoms with different types embedding into the structure of N-GLFs deposited at different synthesis temperatures.

Synthesis Temperature, °C		800	850	900	950	1000
N/C, %		2.1		2.0		2.8
N configuration ^2^	Pyrrolic, %	13.8	9.2	10	4	2.1
	Pyridinic, %	7.3	11.3	4.1	1.8	<0.005
	Graphitic, %	78.9	79.5	85.9	93.8	97.9

^1^ The data reproduced with permission from [23]. ^2^
Appendix A contains deconvoluted X-ray photoelectron spectra (XPS) from which percentage data were extracted.

**Table 2 materials-16-04007-t002:** The capacitance data obtained in literature for graphene-based materials with different thicknesses.

Material ^1^	Method	Electrolyte ^1^	C, mF/cm^2^	Thickness	Ref.
G-GQDs	Electrophoretic deposition	PVA/H_3_PO_4_	0.009	-	[28]
G/EC	Inkjet printing	PVA/H_3_PO_4_	0.028	40 nm	[29]
Graphene	Plasma etching	PVA/H_2_SO_4_	0.081	15 nm	[30]
Graphene	Self-aligned printing	Ionic liquid	0.268	300 nm	[31]
rGO	Self-assembly	PVA/H_3_PO_4_	0.394	-	[32]
rGO	Electrophoretic deposition	PVA/H_3_PO_4_	0.462	25 nm	[33]
rGO/GO/rGO	Direct laser writing	1 M Na_2_SO_4_	0.51	≈25 μm	[34]
SG	Plasma etching	PVA/H_2_SO_4_	0.582	10 nm	[35]
EEG aerogel	Inkjet printing	PSSH	0.7	0.7 μm	[36]
rGO	Self-assembly	PVA/H_2_SO_4_	0.95	27 μm	[37]
*N-GLFs*	*Direct CVD*	*0.5 M Na_2_SO_4_*	*0.96*	*50 nm*	*This work*
WJW-G+SWCNTs	Screen printing	PVA/H_3_PO_4_	1.324	27 μm	[38]
PRG	Photo reduction	PVA/H_2_SO_4_	1.5	2–4.5 μm	[40]
VGN	CVD	Ionic liquid	2	1–2 μm	[39]
rGO	Laser writing	PVA/H_2_SO_4_	2.32	≈8 μm	[41]
N-rGO	Inkjet printing	PVA/H_3_PO_4_	3.4	10 μm	[42]
G/CNTs	CVD	Ionic liquid	3.93	tens μm	[43]
LIG	Laser irradiation	1 M H_2_SO_4_	4	25 μm	[44]
EEG	Spray coating	PVA/H_3_PO_4_	5.4	2 μm	[45]
N-rGO	Spray coating	1M H_2_SO_4_	9.5	700 nm	[46]
PPyG	-	PVA/H_2_SO_4_	22	20 μm	[47]
LIG/MnO_2_	Laser irradiation	0.5 M Na_2_SO_4_	128	-	[48]

^1^ The abbreviations used: G-GQDs—graphene-graphene quantum dots; G/EC—graphene/ethylcellulose; rGO—reduced graphene oxide; SG—sulfur-doped graphene; EEG—electrochemically exfoliated graphene; WJW-G + SWCNTs—wet-jet-milling graphene + single-wall carbon nanotubes; PRG—photoreduced graphene oxide; VGN—vertically oriented graphene nanosheets; N-rGO—N-doped reduced graphene oxide; G/CNTs—graphene on carbon nanotubes; LIG—laser-induced graphene; PPyG—polypyrrole/graphene; PVA—polyvinyl alcohol gel; PSSH—mixing 1 mL of poly(4-styrenesulfonic acid) solution with, in sequence, 0.5 mL of deionized water, 0.5 mL of ethylene glycol, and 0.14 mL of phosphoric acid.

## Data Availability

Not applicable.

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
