# Peer review of "N-Doped Graphene-like Film/Silicon Structures as Micro-Capacitor Electrodes"

_materials, 2023, doi:10.3390/ma16114007_

Round 1

Reviewer 1 Report

In this manuscript, capacitive characteristics of the Nitrogen-doped Graphene-like films on Silicon plates were investigated by CV, EIS, AFM, Raman to be used as microcapacitor electrodes.

Some weak points of the manuscript are given below:

- The literature should be checked for the past year and cited in this manuscript.

- Add the reference to the Equation (1), and the comments that are not derived from this manuscript need references where mentioned (e.g., Lines 170-171, or like the Lines 259-261).

- Distortions in the Nyquist plot of the gray sample (26 hours) should be explained more, full EIS spectra could be shared in Figure 3(a).

- More information could have been given about the adhesion. Did you measure the adhesion quantity?

- In Line 263: It may be better to say "In references" or "in literature" rather than "...by different authors ...". 

- All figures could be expressed with different symbols and arranged in a way suitable for black and white print/output.

- There are typing errors (e.g., Line 257: ... tensof ....).

Quality of English language is not bad. Minor editing of English language required.

Author Response

I thank referee for the valuable comments. Changes in the text are highlighted in green unless otherwise specified. Point-by-point response is below.

– The introduction is supplemented with new references, published in 2022-2023 (ref 3, 5, 6, 11, yellow-marked).

– I added the reference 22 (yellow-marked) to the Equation (1), as well as ref 23 in lines 172, 182 and ref 28-48 in line 271.

– I added the explanation in lines 224-225. Figure 3a shows the full EIS spectra in the measured frequency range (50 kHz–10 Hz).

– Unfortunately, I did not measure adhesion quantity. I only assume the worst adhesion at the lowest synthesis temperature due to extreme capacity loss during cycling of 800 °C-sample.

–  Agree. I corrected this omission.

–  I did it.

–  I fixed it.

Reviewer 2 Report

The authors have proposed a simple and efficient approach for the transfer-free CVD synthesis of N-doped graphene-like films (N-GLFs) on nonmetals by pyrolysis of acetonitrile. The electrochemical properties of the N-GLFs synthesized directly on silicon investigated and their energy storage behavior described, taking into account the bonding configurations of nitrogen and the thickness of the N-GLFs.

1) Equation 1 should be verified with Ref.   
2) The conclusions should be improved and reformulated.  
3)The CVD process should be explained in detail.

Author Response

–  I appreciate the valuable comments given to me by the Reviewer.  I have made all suggested corrections. See lines:

1) 116;

2) 324-341;

3) 93-97.

Reviewer 3 Report

In this manuscript “N-doped graphene-like film/silicon structures as micro-capacitor electrodes” authors did interesting work in very well way; however, it needs major corrections/ additions before further proceedings.

1.     Change “microcapasitor” from the title to “micro-capacitor”.

Your main work is related to doped graphene so add 1 paragraph in introduction about graphene, it’s properties and applications etc. Look this article (DOI: 10.3390/nano12213745) and cite them in line 28. Also, Cite this article (doi.org/10.1002/adfm.202205600) in line 41.

2.     In section 2.1 N-GLF Deposition
From where you purchased silicon plates? mention them properly
Did you clean or pre-treat your substrate before coating? If yes, then provide proper details. It is very important that the roughness or activation of substrate etc. before coating as it play a vital role during deposition/ coating process.
Provide the list or make a proper table about the used parameters during coatings (time, temperature, pressures etc)

3.     In section 2.2 “Measurement”
You placed your coated sample/ electrode in electrolyte for overnight. What was the total impact for you doing this? Provide CV curves without doing this for better comparison. It might be an interesting point here.

4.     You mentioned about GCD measurement at page# 3 line# 100 to 101. However, you did not provide the data of GCD in this study
Add GCD curves of your tested samples after CV curves data, calculate specific capacitance through GCD curves and discuss them properly. Herein, the data from CV curves is not sufficient.

5.      It is also better to cite these references (doi: 10.1038/s41893-023-01101-z and doi: https://doi.org/10.1016/j.snb.2022.132846) in line 39.

6.     The thickness of coatings plays their role in electrochemical study. You mentioned AFM at page# 3 line# 110 to 111 but did not provide any data in this study or in supplementary data. Although, you provide some data through Raman spectroscopy, but it is not enough.
I advised you to add a proper portion of AFM in this study, only these 2 lines are not sufficient. If possible, then also add cross-sectional FESEM images of your prepared samples and discuss them here properly and corelate these findings with the electrochemical study. As coating thickness and morphology have huge impact on the overall electrochemical properties of the coatings.     

7.     In Fig. 1 at page# 4 use different symbols for (a) and (b) sub figures while, use bar chart for sub figure (c) and (d).

8.     Provide proper reference at the end of line# 150 and start of 151 at page# 4 regarding this paragraph and your conclusion.  

9.     At page# 6, line# 221 to 223. It is confusing, rewrite it with proper time scale etc.

10.  At page# 6, fit your Nyquist plot via using the proper electrochemical cell equivalent circuit and discuss these results herein and corelate them properly with your prepared samples capacitance characteristics.

11.  Prepare a proper device and add its data in this article.  

12.   Improve your conclusion section.

Minor Englished improvement required.

Author Response

– I am very grateful to the Reviewer for the valuable work on my manuscript. When I revised the manuscript (changes in the text are highlighted in yellow) I tried my best to make it more intelligible. I ask the Reviewer to take into account the restricted time for revision and the limited volume of the manuscript. Point-by-point response is below.

  1. I corrected the typo in the title and cited all mentioned articles (ref 3, 5).
  2. I've added information about silicon plates and their pre-treat to lines 73-74 and 75-76, respectively. Because ultrasonic treatment in isopropanol is standard wafer processing, I believe it has no significant effect on deposition process. I provided a proper table about the used parameters during coatings in Supplementary Materials (Table S1). The table was also mentioned in the main text (line 97).
  3. I added appropriate CV curves in SM (Fig. S-2-0) as well as corresponding sentence and relevant reference in lines 102-104.
  4. I added GCD curves and its discussion in SM (Section S5) as well as corresponding sentence in lines 200-202.
  5. I added articles (doi: 10.1038/s41893-023-01101-z) in line 39 (ref 11) and (https://doi.org/10.1016/j.snb.2022.132846) in line 31 (ref 6).
  6. I added AFM image in SM (Fig. S6) as well as corresponding sentence in line 260. Unfortunately, there is no way to make a cross-sectional FESEM in a short time.
  1. I did it.
  2. I added ref. 24 in line 162.
  3. I did it, see line 236.
  4. EIS is certainly a valuable and informative method of analysis. Here, I use it rather crudely and I assume that in this work, the processing of EIS spectra will unreasonably make the text heavier.
  5. This is a complex scientific and technological task which looks unreal in this article.
  6. I improved and reformulated conclusion section.

Round 2

Reviewer 2 Report

It is improved in this version, 

It is improved in this version, 

Reviewer 3 Report

The revised manuscript is acceptable for publication as authors replied to all questions properly.

Minor grammatical corrections are required.